# Repurposing Existing Infrastructure for Urban Air Mobility: A Scenario Analysis in Southern California

**Xiangyu Li**

Department of Urban Planning and Design, School of Architecture, Tsinghua University, Beijing 100084, China; xiangyu_li@tsinghua.edu.cn; Tel.: +86-010-62783496

**Abstract:** The deployment of urban air mobility in built-out metropolitan regions is constrained by infrastructure opportunities, land use, and airspace zoning designations. Meanwhile, the availability and spatial distribution of infrastructure opportunities influence the travel demand that can be potentially captured by UAM services. The purpose of this study is to provide an initial assessment of the infrastructure opportunities of UAM in southern California with different mixes of spatial constraints, such as noise levels, school buffer zones, and airspace zones. The corresponding travel demand that can be potentially captured under each scenario is estimated with a home–workplace trip table. The results of the analyses indicate that supply-side infrastructure opportunities, such as heliports and elevated parking structures, are widely available to accommodate the regional deployment of UAM services. However, current spatial constraints can significantly limit the scope of vertiport location choices. Furthermore, the low-income population, blue-collar workers, and young people live farther away from supply-side opportunities than the general population. Moreover, this study proposes a network of UAM based on the top home-based and workplace-based stations for long-distance trips.

**Keywords:** urban air mobility; vertiport; infrastructure





## 1. Introduction

The transportation sector faces the challenge of meeting the growing demand for convenient passenger mobility while reducing congestion, improving safety, and mitigating emissions. Automated driving and electric propulsion are disruptive technologies that may contribute to these goals, but they are still limited by congestion on existing roadways and land-use constraints. Urban air mobility (UAM) can positively contribute to a multimodal mobility system by leveraging the sky to better link people to cities and regions. Commercial UAM operations have begun in the United States since at least the 1940s. For example, from 1947 to 1971, Los Angeles Airways used helicopters to transport people and mail between dozens of locations in the Los Angeles basin, including Disneyland and Los Angeles International Airport [1,2]. During the same era, from 1949 to 1979, New York Airways primarily used helicopters to fly people between helipads in Manhattan and airports in the New York area, such as LaGuardia, JFK, and Newark. Tragically, these initial commercial experiments experienced several fatal accidents. After those accidents, the once-booming operation of helicopter-based UAM was halted nationwide, and the resulting financial difficulties led these companies to cease operations [3]. Despite these historical operation failures, recent technological advances provide an opportunity for the resurgence of urban air mobility.

According to NASA [4], the electric vertical takeoff and landing (eVTOL) UAM system has reached "a level of maturity to enable UAM using safe, quiet, and efficient unmanned vehicles to conduct on-demand and scheduled operations (p. 1)". Types of UAM operations could be emergency responses, humanitarian missions, newsgathering, package delivery, and passenger transport. Among these potential applications, the use of passenger transport is of the greatest attention given its promising future to alleviate traffic congestion

with green technology at a new dimension. For instance, a study in the Bay Area has provided evidence that the UAM system would have a significant impact on reducing the trip time for trips greater than 15 miles [5]. In addition, fully loaded eVTOL's greenhouse gas emissions per passenger-kilometer are less than half of internal combustion engine vehicles and 6% lower than ground electric vehicles [6].

Despite the technological advances and growing commercial interest, it is important to note that the adoption of UAM still faces multiple challenges, as highlighted by many researchers. Such challenges include estimating the demand for air taxi services (transportation modeling and market analysis), air traffic control, operation, infrastructure planning, safety, and regulations [4,7–10]. One of the most critical issues for scaling UAM in built-out metropolitan regions is identifying appropriate landing sites along with other supply-side opportunities and constraints, such as no-fly zones, noise levels, and school zones, to accommodate the regional-wide deployment of eVTOLs [11–13].

To identify the infrastructure opportunities and assess the impact of spatial constraints on these opportunities, this study (1) explores the availability and spatial distribution of current urban infrastructure that can be potentially used as UAM landing sites or vertiports; (2) conducts scenario analyses on how various spatial constraints affect the infrastructure opportunities of UAM; and (3) investigates how different scenarios affect the accessibility of vertiports for different groups of home–workplace commuters.

The paper proceeds as follows: Section 2 presents the literature review. Section 3 describes the data collection process and scenario analysis framework. Section 4 presents the results of the scenario analysis, and Section 5 discusses the conclusion, implications, and limitations of this study.

## 2. Literature Review

Although urban air mobility (UAM) is an emerging mode of transportation, many studies have been conducted to understand the factors associated with transportation technology acceptance. In particular, the social acceptance and user perceptions of ground autonomous vehicles (GAVs) have received increasing scholarly attention in recent years. In the case of UAM, examining recurring factors in relevant studies of ground autonomous vehicles and on-demand aerial service provides valuable insights into the future of UAM adoption, given that only a few pioneering studies have focused on users' preferences for UAM.

### 2.1. Demand-Side Factors Associated with UAM or GAV Adoption

Increasing scholarly and institutional attention has been paid to understanding factors associated with the adoption of autonomous transportation technologies. In recent years, social barriers and key factors related to user adoption have been widely studied for ground autonomous vehicles. Although GAVs are different from UAM from a technological perspective, they share many characteristics in user adoption from the innovation diffusion perspective. Sociodemographic factors, such as income level and gender, were significantly associated with autonomous vehicle acceptance [14,15]. Higher-income, technology-savvy males who live in urban areas and those who have experienced more car accidents (risk takers) have a greater interest in and higher willingness to pay for a GAV [14]. User choice is also influenced by social networks, including neighbors and close friends. Ref. [16] pointed out that users' perceptions of risks hinder the adoption of GAVs (e.g., data privacy and remote hacking). While trust is the most critical factor related to GAV acceptance, perceived ease of use and perceived usefulness are also significantly related to innovation adoption [17]. While GAVs hold the promise of reducing the number of crashes and fatalities on the roads, safety concerns are one of the major barriers to promoting the adoption of GAVs [18].

Interestingly, these concerns about GAV adoption reappear in early studies on UAM adoption. In a stated preference survey study in Germany, ref. [19] explored user perception on UAM. The study indicated that safety and trust are primary concerns for UAM adoption,

and adopters are younger and have higher incomes. Ref. [20] extended the study of UAM adoption to Munich, Germany, and their results further suggested that younger individuals and older populations with higher household incomes are more likely to adopt UAM. Moreover, trip purpose proved to be a significant consideration, with noncommuting travel being the respondents' most preferred GAV option. Airbus's survey [21] in four different countries reported that communities are most concerned about safety followed by the type of sound generated from the aircraft and then the volume of sound from the aircraft. Less than half (44%) of all respondents' initial reactions to UAM are in support or strong support while 41% of all respondents believe UAM is either safe or very safe. Deloitte's report provided similar insights that demographic factors are significantly associated with UAM adoption as younger generations (Gen Y and Gen Z) are more likely to agree that UAM provides an efficient alternative mode of urban transportation [22]. Table 1 presents a summary of recent studies on factors associated with UAM or AV adoption.

**Table 1.** Recent demand-side key studies on UAM adoption.

| Authors | Year of Publication | Area of Research | Study Area | Key Findings |
|---|---|---|---|---|
| Al Haddad et al. | 2020 | UAM | Germany | (1) Safety and trust, affinity to automation, data concerns, social attitude, and sociodemographics are important issues for UAM adoption; (2) skeptical respondents had behavior similar to late and nonadopters [19]. |
| Bansal et al. | 2016 | GAV | Austin, Texas | (1) Higher-income, technology-savvy males who live in urban areas and those who have experienced more crashes have a greater interest in and higher willingness to pay for AV; (2) user choice is dependent on friends' and neighbors' adoption rates [14]. |
| Fu et al. | 2019 | UAM | Munich, Germany | (1) Travel cost and safety may be critical determinants in UAM adoption; (2) younger individuals as well as older individuals with high household income are more likely to adopt UAM; 3) during the market entry stage, potential travelers may favor UAM particularly for performing noncommuting (recreational flying) trips [20]. |
| Roy et al. | 2019 | UAM | U.S. (CSAs) | (1) Near-term eVTOL aircrafts may be able to dramatically increase the expected user base compared to present-day helicopters flying the same mission; (2) assumptions including vehicle design range, payload, and cruising speed can change the results significantly [21]. |
| Hohenberger et al. | 2016 | GAV | Germany | (1) The anxiety level has a significant gender-related difference towards AV adoption; (2) differential effect of sex on anxiety was more pronounced among relatively young respondents and decreased with participants' age [15]. |
| Kyriakidis et al. | 2015 | GAV | 109 countries | (1) Most people think manual driving is still the most enjoyable mode of travel; (2) respondents were found to be most concerned about software hacking/misuse and were also concerned about legal issues and safety; (3) respondents from more developed countries were less comfortable with their vehicle transmitting data [16]. |
| Lidynia et al. | 2016 | Drones | Aachen University, Germany | (1) Laypeople feared the violation of their privacy whereas active drone pilots saw more of a risk of possible accidents; (2) participants had clear expectations regarding the routes drones should and should not be allowed to use [23]. |

| Authors | Year of Publication | Area of Research | Study Area | Key Findings |
|---|---|---|---|---|
| Deloitte | 2019 | UAM | 20 countries | (1) Despite the substantial progress made in terms of vehicle design and technology, consumers continue to doubt the safety of UAM; (2) regional and generational differences play a critical role in the perceived UAM safety; (3) 49% of respondents in the United States were unconvinced about UAM safety while only 39% are skeptical about safety in China; (4) younger consumers surveyed (Gen Y and Gen Z) agree that UAM provides for an efficient alternative mode of urban transportation but are more apprehensive about its safety [22]. |
| Peeta et al. | 2008 | On-demand air service (ODAS) | Indiana, Illinois, and Florida | (1) Travel distance, service fare, and the ODAS location are key factors influencing user switching decisions (from traditional air service to ODAS); (2) ODAS landing location is a key determinant of its operational viability and has significant implications for policymakers, regional/city planners, operators, and businesses; (3) while ODAS proximity to work is an attractive attribute for users, environmental concerns can arise if close to residential areas [24]. |
| Airbus | 2019 | UAM | Los Angeles, Mexico City, New Zealand, and Switzerland | (1) Communities are most concerned about safety followed by the type of sound generated from the aircraft and then the volume of sound from the aircraft; (2) other concerns include the time of day at which aircrafts are flown and the altitude at which aircraft fly; (3) 44.5% of all respondents' initial reactions to UAM is in support or strong support while 41.4% of all respondents believe UAM is either safe or very safe [21]. |
| Zhang et al. | 2019 | GAV | China | (1) Trust was the most critical factor in promoting AV acceptance; (2) perceived ease of use (PEOU) and perceived usefulness (PU) were significant factors; (3) the effects of PEOU and PU were weaker compared to trust [17]. |

### 2.2. Supply-Side Factors Related to UAM Adoption

While demand-side factors have attracted lots of scholarly attention in early studies on UAM, the identification of viable UAM landing sites and other supply-side constraints/opportunities are preconditions to advance empirical studies on UAM adoption. Uber, Airbus, and other pioneers in the UAM industry have provided their archetypes of UAM infrastructure, some of which are based on existing urban infrastructure, such as elevated parking structures, parking lots, and airport terminals (Figure 1). Recent studies point out that early UAM vertiports can leverage existing airport infrastructure given the high customer willingness to pay and substantial time-savings for long-distance trips [25]. Refs. [26,27] suggested that vertiports can leverage underutilized urban infrastructures, such as helicopter pads, barges over water, inside highway cloverleaves, and qualified rooftops (e.g., parking structures), with the constraints of other supply-side, such as air space zones, land use regulations, and population density. These findings coincide with early studies (e.g., see [24]) on on-demand aerial service (ODAS), which suggested that landing location is a key determinant of its operational viability. While an ODAS's proximity to work is an attractive attribute for users, environmental concerns can arise if it is close to residential areas. Ref. [27] emphasized that median income distribution, land value, and job density should be considered in the placement of UAM landing sites. Ref. [28] identified another research trajectory in which UAM vertiports served as a transition hub for air travelers between long-distance air travel and short/medium-distance air travel. In

his work, vertiport placement was optimized based on airport traveler data. Moreover, the operational capacities of ground infrastructure play an important role in improving the operation efficiency of UAM systems. Ref. [26] suggests that balancing the number of gates and the number of vehicles will maximize the utility of UAM mode when vehicle specifications are held constant.

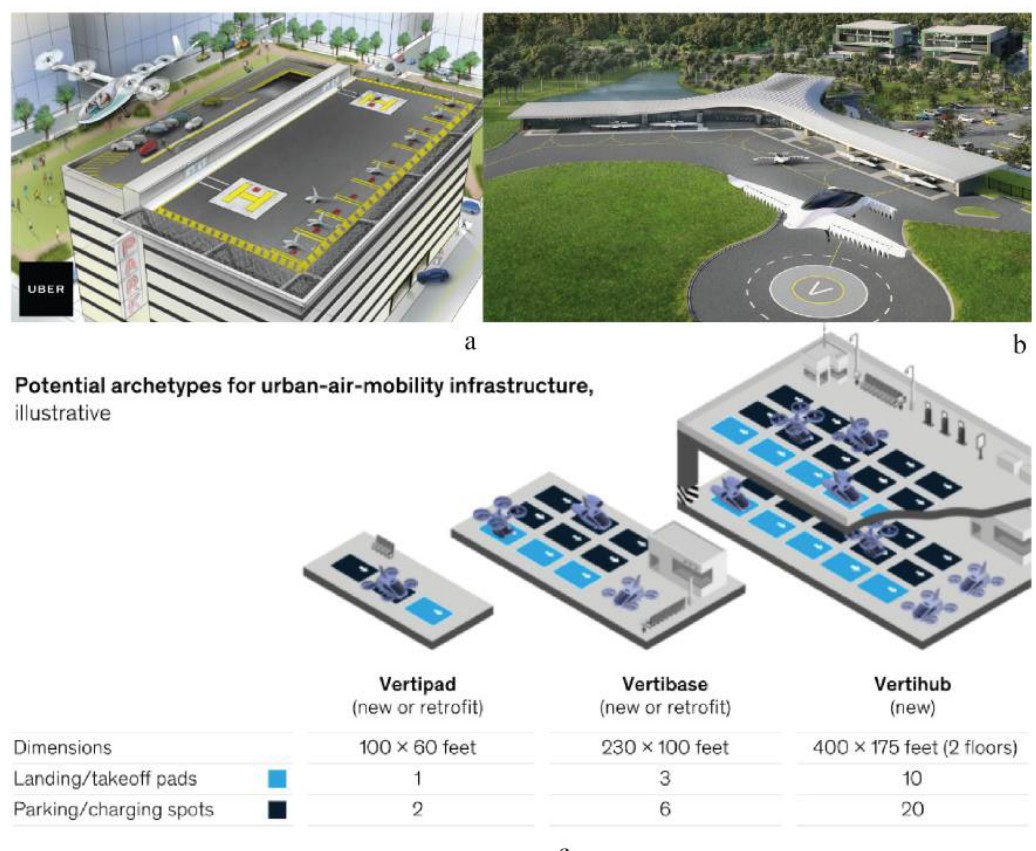

**Figure 1.** Examples of UAM landing sites. (**a**) UAM landing site design by repurposing underutilized parking structure rooftops [29]. (**b**) Lilium UAM vertiport in Lake Nona, Florida [30]. (**c**) Archetypes of UAM infrastructure [31].

## 3. Data and Analysis

### 3.1. Study Area

This study focuses on the Greater Los Angeles Area, which is composed of five populous counties (Los Angeles County, Orange County, San Bernardino County, Riverside County, and Ventura County). Supply-side infrastructure opportunities, such as heliports and elevated parking structures, are widely available to accommodate the regional deployment of UAM services (see Figure 2). Additionally, there is an increasing local interest in UAM adoption in the study area. Local government plays a vital role in gaining public trust for a new mode of transport, such as UAM, by facilitating conversations between community members and the private sector [32]. In December 2020, Mayor Garcetti announced a public–private partnership between the Los Angeles Department of Transportation and Hyundai Motor Group; this effort aims to introduce UAM to local airspace by 2023 [33]. One of the early goals of this partnership is to visualize a few vertiports where people can go flying on eVTOL. Moreover, the climate of southern California is classified as a Mediterranean climate, a type of dry subtropical climate. Such a climate is desirable for the all-year-round operation of UAM systems. Additionally, on the demand side, this region has seen a significant increase in 'super commuters' whose round trip to work is

more than 3 h. According to the 2018 5-year American Community Survey, there are over 150,000 super commuters, or 1.5% of the total population in Los Angeles County [34].

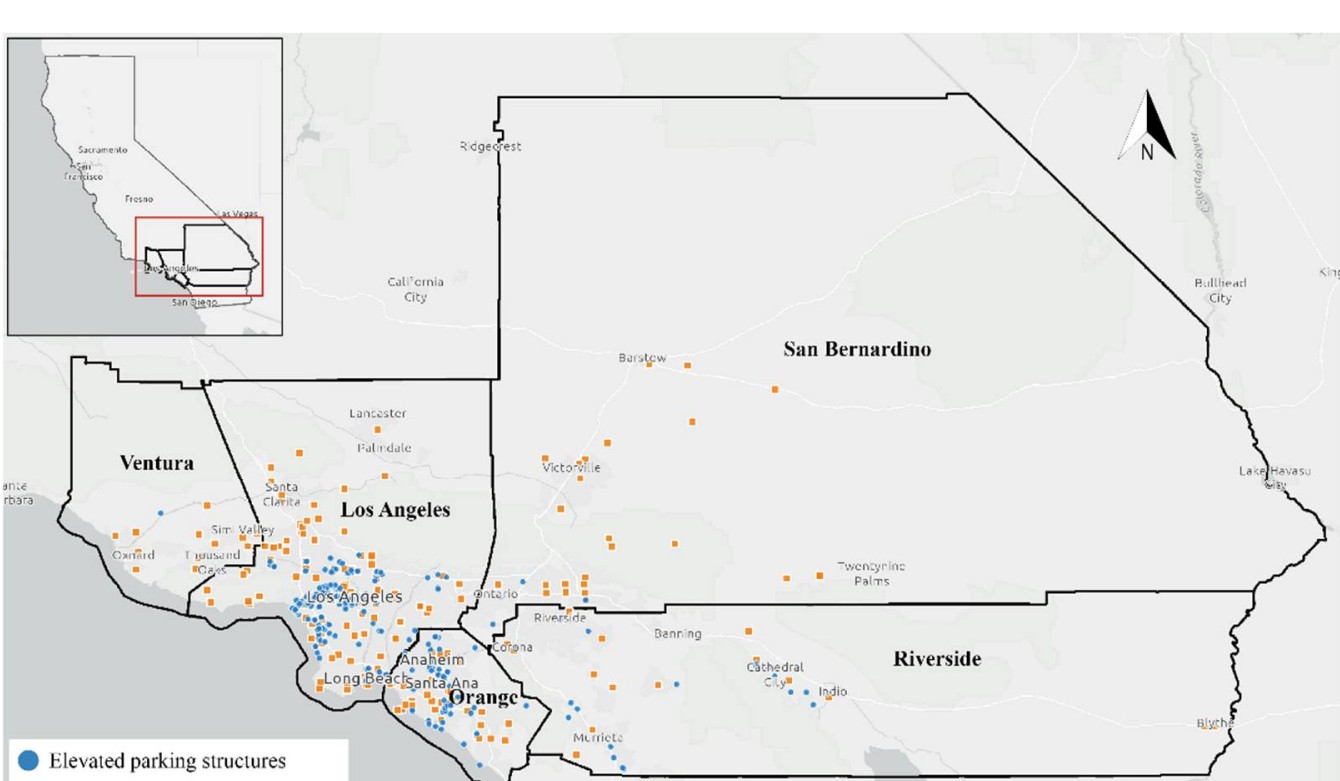

**Figure 2.** Study area.

*3.2. Data*

This study collected two categories of data for the scenario analyses. On the demand side, this study relied on the U.S. Census Longitudinal Employer–Household Dynamics (LEHD) Origin–Destination Employment Statistics that capture home–workplace trips (origin/destination locations) and commuter characteristics at the census block level [12]. The LEHD dataset is based on employer reports of quarterly earnings that integrate workers and employers. Moreover, the LEHD data provided three categories of sociodemographic characteristics associated with commuting trips, including age groups, income levels, and occupations. The data breaks of different commuter characteristics are presented in Table 2 The scenario analysis used the LEHD Origin–Destination (OD) data to estimate the potential travel demand and commuter characteristics under different supply scenarios of vertiports. One of the advantages of LEHD data compared to other trip datasets, e.g., Census Transportation Planning Products (CTPP) sample-based surveys, is their comprehensive and timely OD geographical matrix for flows between households and workplaces at fine-grain that can be used in transportation analysis. However, LEHD does not include mode choices and travel costs information (i.e., costs, time, etc.). Therefore, this study focuses primarily on the accessibility of potential vertiport locations while the available data exclude the possibility of comparing the actual travel costs between UAM and other modes of travel.

On the supply side, this study considered data sources that captured landing site opportunities and spatial constraints. Infrastructure opportunities include helipads and elevated parking structures collected for the study area. The helipad location data was

derived from the FAA's National Airspace System Resource Aeronautical Data Product [35]. The helipad database is a geographic point database of aircraft landing facilities in the United States. Attribute data is provided on the physical and operational characteristics of the landing facility, current usage including enplanements and aircraft operations, congestion levels, and usage categories. The elevated parking structure locations were piled with the Google Map API. Each elevated parking structure location was manually verified with Google Earth satellite images. Moreover, the supply-side constraints, including national airspace zones and local land use constraints (e.g., noise levels and schools) were collected for the analysis. Specifically, the noise data were retrieved from the National Transportation Noise Map, which allows the measurement of potential exposure to aviation and highway noise. No-fly-zone thresholds are defined in accordance with the UAS airspace regulation [36]. Table 2 summarizes the identified factors and corresponding data sources. The school zone datasets were collected from the California School Campus Database (CSCD) [37]. The CSCD dataset includes parcel-level campus boundaries of schools with kindergarten through K-12 instruction as well as colleges, universities, and community colleges. The collected datasets and their sources are presented in Table 2.

**Table 2.** Factors and data sources.

| Categories | Factors | Data Sources | Unit of Observation | Year of Collection |
|---|---|---|---|---|
| Demand side | Age groups | Census Origin–Destination Employment Statistics | Census block | 2019 |
| | Occupation groups | Census Origin–Destination Employment Statistics | Census block | 2019 |
| | Income groups | Census Origin–Destination Employment Statistics | Census block | 2019 |
| | Home–workplace trips | Census Origin–Destination Employment Statistics | Census block | 2019 |
| | Home–workplace distance | Calculated from the centroids of census blocks | Census block | 2019 |
| Supply side | Helipads | The Federal Aviation Administration | Site | 2019 |
| | Parking structure | Google Earth satellite image/Google Map API | Site | 2021 |
| | UAS airspace and facility map | The Federal Aviation Administration | Zone | 2020 |
| | Schools | California School Campus Database | Land Parcel | 2018 |
| | Noise levels | The United States Department of Transportation | - | 2018 |

### 3.3. Scenario Design

To understand how various spatial constraints affect the accessibility of potential vertiport locations among various commuting groups, this study applied different spatial constraint scenarios (see Table 3). This study defined five simple but crucial categories of rules to define demand-side constraints, supply-side constraints, and accessibility of vertiports. All these rules were mixed and applied to the five-county metropolitan area to measure the effects of spatial constraints on vertiports accessibility. The demand-coverage rules and block-distance rules enabled this study to further evaluate the impact of spatial constraints on vertiport location choice by home–work commuter characteristics (e.g., travel distance).

The demand-side rules considered home–workplace commuters by various population characteristics (e.g., age, income, and occupation) defined by the LEHD dataset (n2). Specifically, the rules included the total number of home–workplace commuters in each census block, age groups that divide the population by young (29 or younger), middle-aged (30 to 54), and senior (55 or older), income levels provided by LEHD that highlight income-disadvantaged populations (e.g., earnings of $1250/month or less), and occupation characteristics with a focus on blue-collar employers (e.g., commuters in trade, transportation, and utility industries). The available LEHD breakdowns of work–home commuters tend to highlight the economically disadvantaged population, which might create biases in understanding the impact of spatial constraints on middle-class commuters' accessibility to vertiports.

To evaluate the effects of supply-side constraints on the accessibility to vertiports, this study proposes different sets of airspace, land use, and noise-level constraints. The

airspace constraints use the FAA UAS Facility Map (UASFM), which depicts the maximum altitude in feet above ground level (AGL) that may be assigned by an FAA processor without additional internal FAA coordination. Although most commercial drones follow UASFM guidance in their ground control applications (e.g., no-fly zones and restrictive fly zones), it is unnecessarily applied to future UAM. Therefore, the use of the UASFM in this study serves as a relatively conservative rule when evaluating the impact of airspace constraints on vertiports, as UAM might share more airspace with manned aircrafts in the future than commercial drones. Apart from constraints in the airspace, existing noise levels on the ground also affect the vertiport choice. The noise data were recoded into three thresholds, 85 dB, 70 dB, and 60 dB, according to the noise impact classifications by the Centers for Disease Control and Prevention [38]. Noise at 60 dB is the equivalent sound level of everyday conversation. People may feel annoyed at 70 dB, and levels above 85 dB are damaging to hearing. Furthermore, this study defined accessibility rules by walking and driving time from the centroids of workplace/home census blocks. The access time was calculated with the ArcGIS online application by considering the speed limit and average traffic flows on the roads. Demand coverage and block distance rules were introduced to further evaluate the impact of supply-side constraints on commuters' accessibility to vertiports by home–workplace travel distance. As shown in Figure 3, five categories of supply-side rules and demand-side rules are mixed to generate the statistics of scenario analyses.

**Table 3.** Summary of scenario analysis rules.

| Categories | Rules | Description |
|---|---|---|
| Demand side | s000 | Total number of home–workplace commuters |
| | sa01 | Home–workplace commuters aged 29 or younger |
| | sa02 | Home–workplace commuters aged 30 to 54 |
| | sa03 | Home–workplace commuters aged 55 or older |
| | se01 | Home–workplace commuters with earnings of $1250/month or less |
| | se02 | Home–workplace commuters with earnings of $1251/month to $3333/month |
| | se03 | Home–workplace commuters with earnings greater than $3333/month |
| | si01 | Home–workplace commuters in goods producing industry sectors |
| | si02 | Home–workplace commuters in trade, transportation, and utility industry sectors |
| | si03 | Home–workplace commuters in all other services industry sectors |
| Supply side | A1 | ≤200 ft preapproved UAS fly altitude |
| | A2 | ≤300 ft preapproved UAS fly altitude |
| | A3 | ≤400 ft preapproved UAS fly altitude |
| | S1 | ≥0.1 miles school buffer |
| | S2 | ≥0.25 miles school buffer |
| | S3 | ≥0.5 miles school buffer |
| | N1 | ≥85 dB noise level |
| | N2 | ≥70 dB noise level |
| | N3 | ≥60 dB noise level |
| Accessibility | D1 | ≤3 min walking by network distance |
| | D2 | ≤5 min walking by network distance |
| | D3 | ≤10 min walking by network distance |
| | D4 | ≤3 min driving by network distance |
| | D5 | ≤5 min driving by network distance |
| | D6 | ≤10 min driving by network distance |
| Demand coverage | J1 | Home–workplace commuters w/only home block access |
| | J2 | Home–workplace commuters w/only workplace block access |
| | J3 | Home–workplace commuters w/both home and workplace access |
| Block distance | s000_10 | Block centroid distance ≥10 miles |
| | s000_20 | Block centroid distance ≥20 miles |
| | s000_30 | Block centroid distance ≥30 miles |

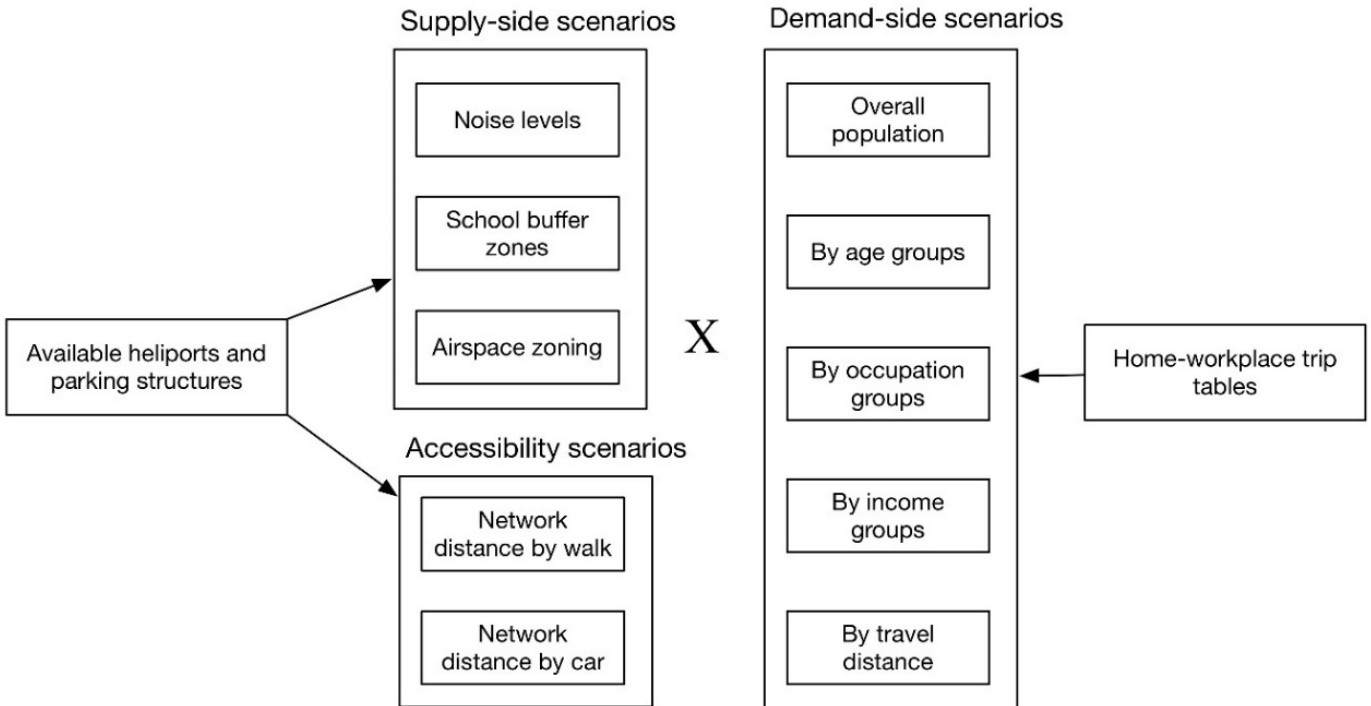

**Figure 3.** Scenario analysis framework.

## 4. Results

The impact of spatial constraints on supply-side opportunities is presented in Appendix A. In general, current highway and airport noise have minimal impact on the location choice of vertiports. At the same time, airspace zoning and school buffer zones can significantly affect the location choice of vertiports. Even under the 200 ft flying altitude restriction, only 67.2% of available vertiports remain valid choices. If the aerial zoning designation keeps the current 400 ft flying altitude (for small drones) in place, only 57.8% of existing infrastructure remain valid, less than half of which are parking structures. By mixing the scenarios, this study has been able to generate 27 (three airspace constraints × three school buffer constraints × three noise level constraints) mixed scenarios. The summary statistics of mixed scenarios are presented in Appendix A. The best, the median, and the strictest mixed scenarios are presented in Table 4. Under the best mixed scenario, 61.9% of available sites remain valid, and slightly less than half are parking structures. The percentage of parking structures drops significantly when the constraints become stricter. Under the strictest mixed scenario, about 1/3 of valid sites are parking structures.

Table 5 presents the home–workplace commuters who can access viable vertiports at their homes and workplaces under three different mixed scenarios. A breakdown of commuters with only home access or workplace access to vertiports is presented in Appendix A. The spatial distribution of valid sites under these mixed scenarios is shown in Figure 3. These results indicate that most vertiports are not within walkable distance from either the centroids of home census blocks or workplace census blocks in the study area. Under the best scenario, 15.8% of commuters can access the vertiports within 5 min driving distance while 70.7% of commuters can access the vertiports with less than 10 min of driving. This unique pattern is likely to make parking structures preferable to helipads when making location choices for vertiports. Please note that the calculation of driving distance has considered the speed limit of road networks but has not considered the live traffic and traffic light waiting time. Therefore, it is reasonable to believe that the results are optimistic estimates.

**Table 4.** Impact of spatial constraints on infrastructure opportunities (single-constraint scenarios and selected mixed-constraint scenarios).

| Rules | Description | Helipads | Parking Structures | Total | % Base |
|---|---|---|---|---|---|
| No constraints | All available sites | 360 | 444 | 804 | 100.0% |
| A1 | ≤200 ft airspace zoning | 271 | 269 | 540 | 67.2% |
| A2 | ≤300 ft airspace zoning | 260 | 242 | 502 | 62.4% |
| A3 | ≤400 ft airspace zoning | 242 | 223 | 465 | 57.8% |
| S1 | >0.1 mile school buffer | 348 | 402 | 750 | 93.3% |
| S2 | >0.25 mile school buffer | 294 | 305 | 599 | 74.5% |
| S3 | >0.5 mile school buffer | 219 | 138 | 357 | 44.4% |
| N1 | ≥85 dB noise | 360 | 444 | 804 | 100.0% |
| N2 | ≥70 dB noise | 347 | 441 | 788 | 98.0% |
| N3 | ≥60 dB noise | 325 | 418 | 743 | 92.4% |
| A1 × S1 × N1 | The best mixed scenario | 262 | 236 | 498 | 61.9% |
| A2 × S2 × N2 | The median mixed scenario | 217 | 148 | 365 | 45.4% |
| A3 × S3 × N3 | The strictest mixed scenario | 154 | 56 | 210 | 26.1% |

**Table 5.** The accessibility of supply-side opportunities for home–workplace commuters (% population with both home and workplace access to vertiports).

| Rules | Description | A1 × S1 × N1 (Best) | A2 × S2 × N2 (Median) | A3 × S3 × N3 (Strictest) |
|---|---|---|---|---|
| D1 | ≤3 min walking distance | 0.13% | 0.06% | 0.00% |
| D2 | ≤5 min walking distance | 0.37% | 0.16% | 0.01% |
| D3 | ≤10 min walking distance | 1.50% | 0.63% | 0.08% |
| D4 | ≤3min driving distance | 4.11% | 1.88% | 0.36% |
| D5 | ≤5 min driving distance | 15.80% | 8.30% | 2.58% |
| D6 | ≤10 min driving distance | 70.73% | 49.48% | 27.49% |

The horizontal histograms under the maps in Figure 4a–c present the spatial coverage of different long-distance commuter groups with both home and workplace access to vertiports under three mixed scenarios. Compared to the population average coverage in Table 5, this study finds that commuter accessibility is not very sensitive to travel distance. In other words, long-distance commuters are likely to enjoy equal access to vertiports as average commuters. This piece of evidence may support the claim that current infrastructures are spatially ready to accommodate a regional network of urban air mobility. However, extreme-low-income and low-income populations have a systematically lower level of accessibility than the population average. Consequently, low-income populations are likely to lag in adopting the UAM commute mode. Blue-collar workers and young commuters also have lower-than-average levels of access to vertiports. Such patterns have remained mostly consistent across different accessibility constraints. Appendix A indicates that certain population groups are systematically disadvantaged in terms of their closeness to vertiports regardless of travel distance or accessibility definitions.

Finally, this study has identified the top home-based and workplace-based vertiport candidates based on eligible parking structures for long-distance commuters in the study area from the above analysis (Figure 5). The top home-based vertiports are located in suburban areas. In contrast, the workplace-based vertiports are situated in popular job centers due to the jobs–housing mismatch in southern California. These potential vertiport candidates represent different strategies that future UAM vertiports might take. For instance, suburban districts have been identified with many similar locations, such as elevated parking structures next to a large shopping center, to be adaptively redesigned as vertiports. Station C represents the opportunities to encourage multimodal travel by incorporating UAM into public transport hubs. Station F demonstrates the opportunities of the UAM travel mode to serve both regular commuters and be part of medical emer-

gency services. Moreover, these vertiports may serve as magnets for planners to revitalize urban places by bringing people, activities, and public places together near UAM hubs (e.g., vertiport-oriented communities).

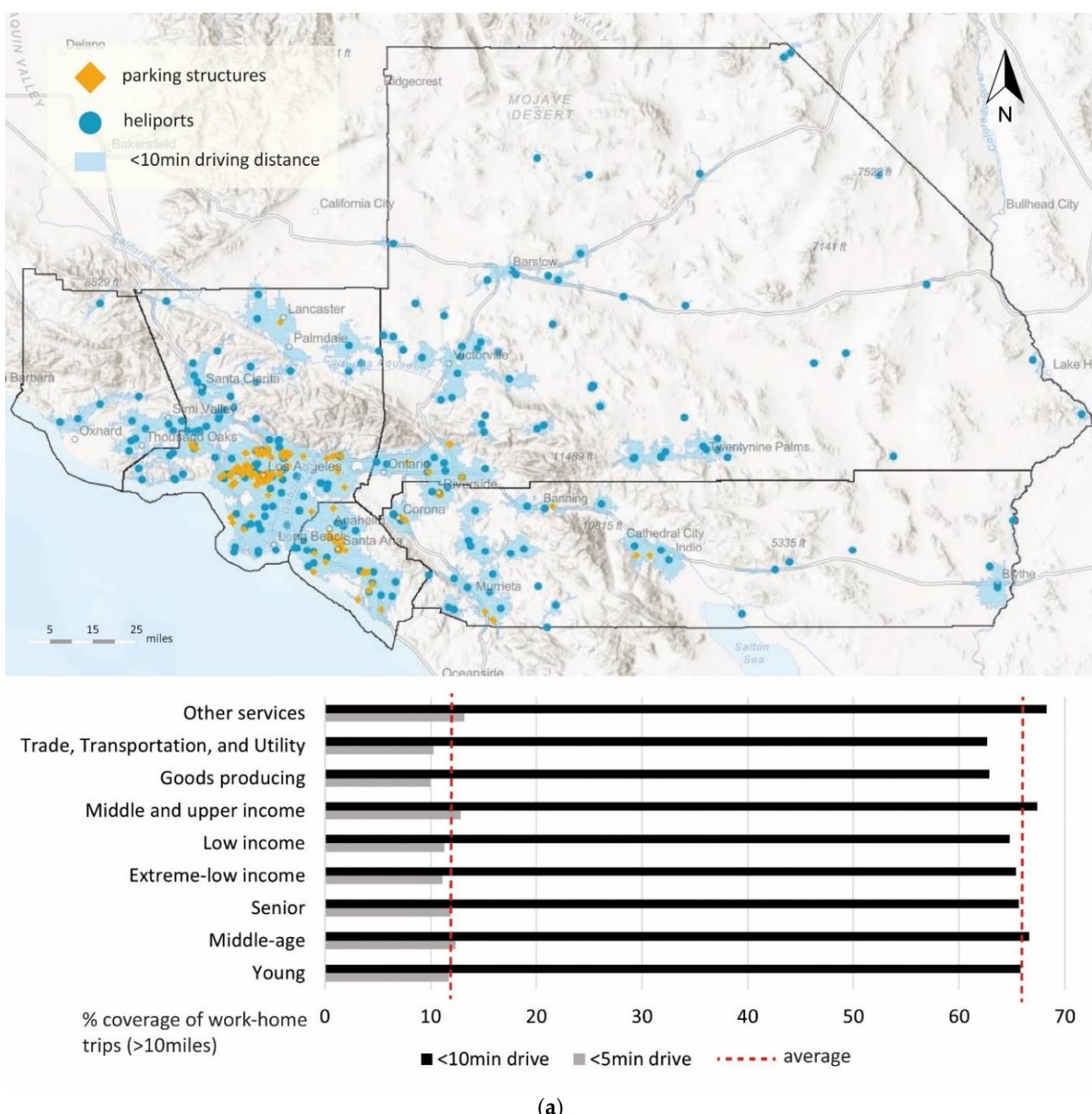

(**a**)

**Figure 4.** *Cont.*

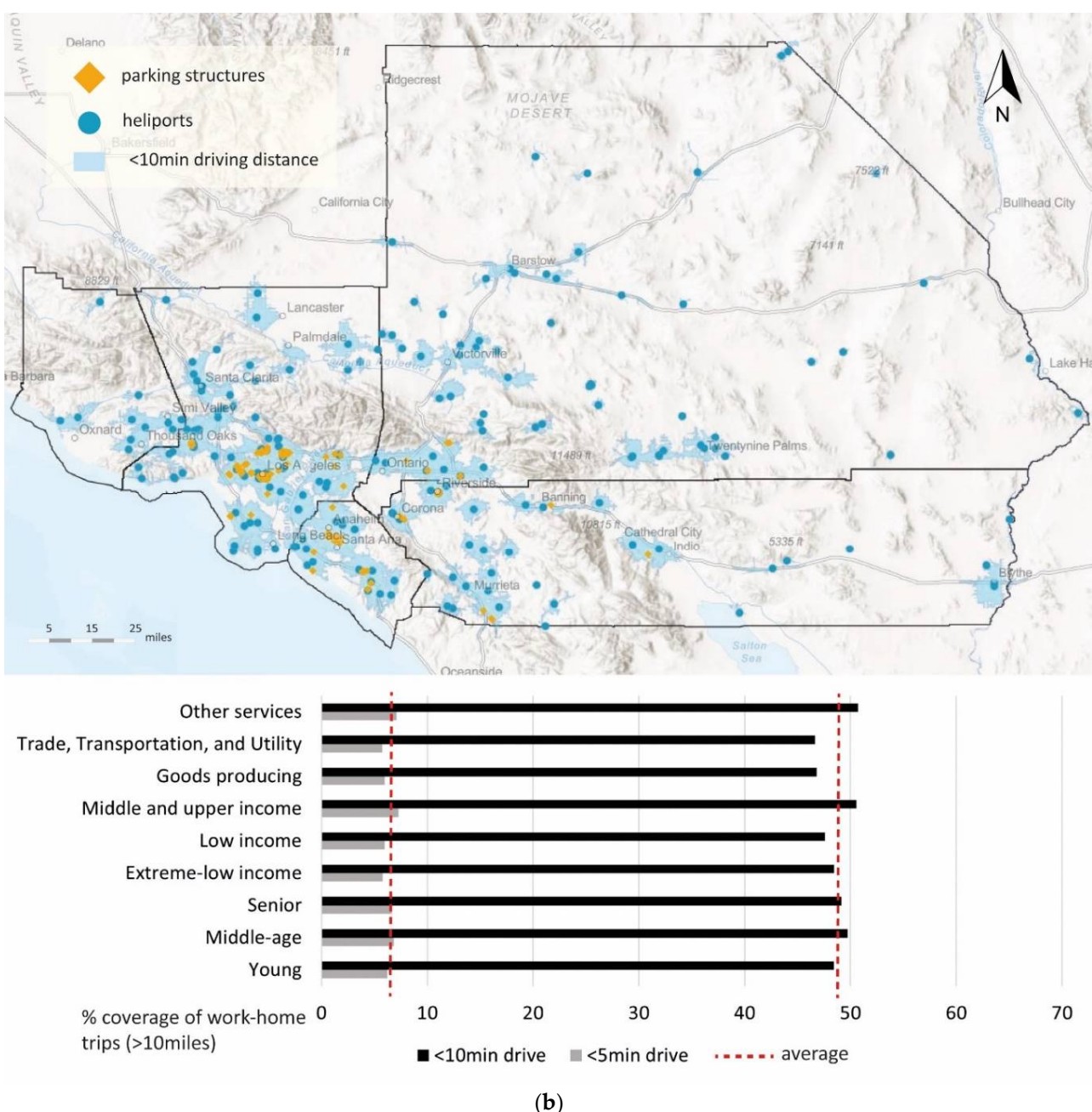

(**b**)

**Figure 4.** *Cont.*

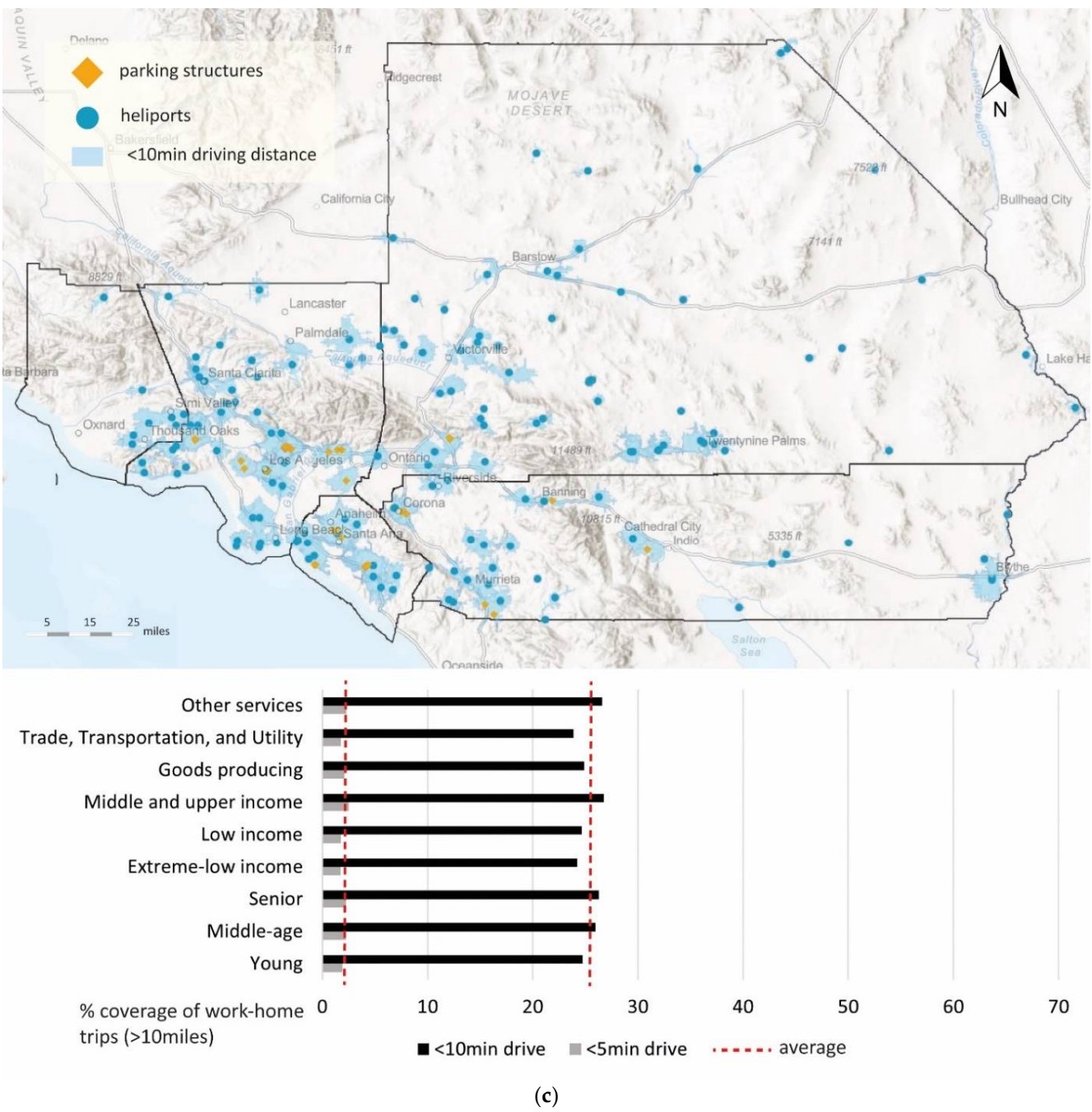

**Figure 4.** (**a**) Spatial distribution of supply-side opportunities and demand-side coverage by user groups based on the best scenario. (**b**) Spatial distribution of supply-side opportunities and demand-side coverage by user groups based on the median scenario. (**c**) Spatial distribution of supply-side opportunities and demand-side coverage by user groups based on the strictest scenario.

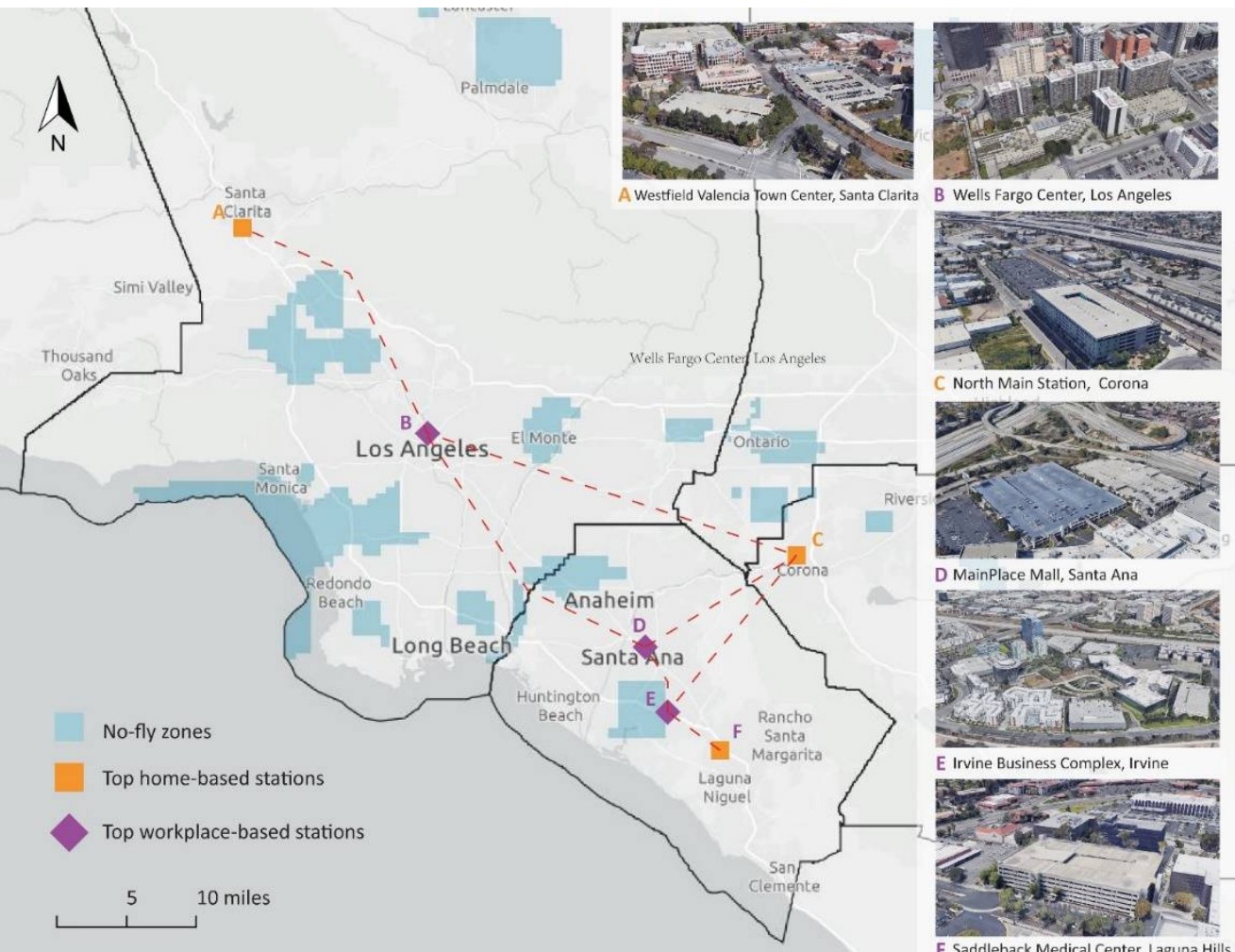

**Figure 5.** A proposed UAM network based on the top home-based and workplace-based stations for residents having commuting trips over 10 miles and living within 10 min driving distance of vertiports.

## 5. Conclusions and Discussion

Urban air mobility holds the promise of becoming a greener, faster, and quieter mode of aerial transportation in the near future [4]. While the adoption of UAM faces many social acceptance barriers [19,38], identifying feasible landing site locations in built-out metropolitan areas remains a major physical barrier to the mass deployment of UAM. This study provides an initial assessment of potential vertiports locations and associated travel demand in southern California by employing a systematic scenario analysis. This study suggests that even under the best scenario, most vertiport locations are not within walkable distance for home–workplace commuters. This pattern is rooted in the urban development patterns of the study area but will make parks and rides a necessary strategy to access UAM in the future. As a result, parking structures, which are already equipped with parking capacity, become a preferable choice of infrastructure to accommodate the future deployment of UAM services. This study also suggests that extremely low-income populations and blue-collar workers have lower accessibility to vertiports regardless of travel distance. Therefore, future policy interventions might be needed for equitable access to this new mode of transportation. This study also extends the analysis by proposing a network of UAM stations in the study area. The illustration provides possible strategies for planners to identify feasible UAM stations that can facilitate the mass adoption of UAM.

Admittedly, this study has several limitations. For instance, supply-side opportunities are not equal to supply-side capacity. The present study does not delve into the utility aspects (e.g., timesaving and costs) of the location choices of UAM vertiports but primarily focuses on supply-side constraints, such as noise levels, school zones, and no-fly zones, and how various user groups might have disproportional access to available vertiports. As briefly discussed in the conclusion section, a major challenge to studying the utility aspects of UAM vertiports placement is the lack of UAM operation data (e.g., ingress/egress time, charging cycles, scheduling, aircraft specification, etc.) [39]. Furthermore, to address the limitation of existing studies, future research might explore the integration of UAM and its impact on cities in the following three major research trajectories: (1) conducting utility-based studies by integrating UAM operation data; (2) conducting multimodal transportation modeling by including UAM operation specifications and vertiports network design; and (3) integrating user adoption factors (e.g., elasticity between user characteristics and UAM demand) in UAM transportation modeling.

Unlike traditional aviation, which by design typically spends the majority of flight time over sparsely populated areas, UAM operations will generally occur over metropolitan areas that are densely populated in terms of people and property. As such, UAM concepts, technologies, and procedures must be designed and managed with safety in mind from the start [40]. As with any new entrant to the airspace, UAM aircraft and operations should be designed in a way to earn acceptance by the public [41]. Additionally, several barriers must be overcome beyond locational constraints for UAM operations to be integrated safely and efficiently into the urban airspace system. The obstacles that are more closely associated with UAM vehicles include ride quality, lifecycle emissions, ease of certification in terms of both time and cost, visual and noise nuisance perceived by the community on the ground, affordability in terms of operating cost, safety in terms of casualties and property damage, and efficiency in terms of energy usage. Concerns about potential privacy violations, auditory and visual disturbances, safety risks, and affordability are some of the significant factors that should be carefully investigated.

**Funding:** This research received no external funding.

**Data Availability Statement:** Not applicable.

**Conflicts of Interest:** The author declares no conflict of interest.

## Appendix A  Summary Statistics of UAM Scenario Analysis

**Table A1.** Impact of spatial constraints on supply-side opportunities (mixed-constraint scenarios).

| ID | Scenarios (3 × 3 × 3 = 27) | Helipads | Parking Structures | Total | % Base |
|----|----------------------------|----------|--------------------|-------|--------|
| 1 | A1*S1*N1 (the best) | 262 | 236 | 498 | 61.9% |
| 2 | A1*S1*N2 | 261 | 234 | 495 | 61.6% |
| 3 | A1*S1*N3 | 250 | 227 | 477 | 59.3% |
| 4 | A1*S2*N1 | 222 | 163 | 385 | 47.9% |
| 5 | A1*S2*N2 | 221 | 162 | 383 | 47.6% |
| 6 | A1*S2*N3 | 214 | 159 | 373 | 46.4% |
| 7 | A1*S3*N1 | 170 | 66 | 236 | 29.4% |
| 8 | A1*S3*N2 | 169 | 66 | 235 | 29.2% |
| 9 | A1*S3*N3 | 167 | 65 | 232 | 28.9% |
| 10 | A2*S1*N1 | 251 | 210 | 461 | 57.3% |
| 11 | A2*S1*N2 | 250 | 208 | 458 | 57.0% |
| 12 | A2*S1*N3 | 243 | 202 | 445 | 55.3% |
| 13 | A2*S2*N1 | 218 | 149 | 367 | 45.6% |
| 14 | A2*S2*N2 (the median) | 217 | 148 | 365 | 45.4% |
| 15 | A2*S2*N3 | 210 | 145 | 355 | 44.2% |
| 16 | A2*S3*N1 | 169 | 60 | 229 | 28.5% |
| 17 | A2*S3*N2 | 168 | 60 | 228 | 28.4% |

**Table A1.** *Cont.*

| ID | Scenarios (3 × 3 × 3 = 27) | Helipads | Parking Structures | Total | % Base |
|----|----------------------------|----------|---------------------|-------|--------|
| 18 | A2*S3*N3 | 154 | 56 | 210 | 26.1% |
| 19 | A3*S1*N1 | 233 | 192 | 425 | 52.9% |
| 20 | A3*S1*N2 | 232 | 190 | 422 | 52.5% |
| 21 | A3*S1*N3 | 227 | 184 | 411 | 51.1% |
| 22 | A3*S2*N1 | 201 | 135 | 336 | 41.8% |
| 23 | A3*S2*N2 | 200 | 134 | 334 | 41.5% |
| 24 | A3*S2*N3 | 195 | 131 | 326 | 40.5% |
| 25 | A3*S3*N1 | 157 | 57 | 214 | 26.6% |
| 26 | A3*S3*N2 | 156 | 57 | 213 | 26.5% |
| 27 | A3*S3*N3 (the strictest) | 154 | 56 | 210 | 26.1% |

**Table A2.** Accessibility of supply-side opportunities for long-distance ($\geq$10 miles) commuters.

| Supply-A1*S1*N1 (Best) | Home Access Only | Workplace Access Only | Both Home and Workplace Access | Home Access/Workplace Access Ratio |
|------------------------|------------------|------------------------|--------------------------------|-------------------------------------|
| $\leq$3 min walking distance | 0.48% | 8.35% | 0.03% | 5.72% |
| $\leq$5 min walking distance | 1.00% | 11.66% | 0.11% | 8.58% |
| $\leq$10 min walking distance | 2.81% | 17.43% | 0.57% | 16.14% |
| $\leq$3 min driving distance | 6.57% | 23.09% | 2.24% | 28.45% |
| $\leq$5 min driving distance | 15.10% | 30.69% | 11.68% | 49.20% |
| $\leq$10 min driving distance | 10.93% | 18.34% | 64.02% | 59.59% |
| **Supply-A2*S2*N2 (Median)** | **Home Access Only** | **Workplace Access Only** | **Both Home and Workplace Access** | **Home Access/Workplace Access Ratio** |
| $\leq$3 min walking distance | 0.30% | 6.47% | 0.01% | 4.70% |
| $\leq$5 min walking distance | 0.64% | 8.99% | 0.05% | 7.12% |
| $\leq$10 min walking distance | 1.79% | 13.12% | 0.23% | 13.64% |
| $\leq$3 min driving distance | 4.70% | 18.02% | 1.03% | 26.05% |
| $\leq$5 min driving distance | 12.57% | 27.16% | 6.41% | 46.27% |
| $\leq$10 min driving distance | 16.57% | 24.38% | 47.61% | 67.94% |
| **Supply-A3*S3*N3 (Strictest)** | **Home Access Only** | **Workplace Access Only** | **Both Home and Workplace Access** | **Home Access/Workplace Access Ratio** |
| $\leq$3 min walking distance | 0.09% | 1.64% | 0.00% | 5.26% |
| $\leq$5 min walking distance | 0.22% | 2.73% | 0.01% | 7.92% |
| $\leq$10 min walking distance | 0.66% | 6.04% | 0.04% | 10.90% |
| $\leq$3 min driving distance | 2.17% | 9.85% | 0.23% | 22.01% |
| $\leq$5 min driving distance | 7.86% | 18.74% | 2.07% | 41.94% |
| $\leq$10 min driving distance | 22.12% | 27.39% | 24.87% | 80.76% |

**Table A3.** Accessibility of supply-side opportunities for long-distance ($\geq$20 miles) commuters.

| Supply-A1*S1*N1 (Best) | Home Access Only | Workplace Access Only | Both Home and Workplace Access | Home Access/Workplace Access Ratio |
|------------------------|------------------|------------------------|--------------------------------|-------------------------------------|
| $\leq$3 min walking distance | 0.40% | 7.85% | 0.02% | 5.05% |
| $\leq$5 min walking distance | 0.84% | 10.92% | 0.08% | 7.71% |
| $\leq$10 min walking distance | 2.43% | 16.64% | 0.42% | 14.61% |
| $\leq$3 min driving distance | 6.00% | 22.36% | 1.92% | 26.85% |
| $\leq$5 min driving distance | 14.45% | 30.88% | 10.65% | 46.79% |
| $\leq$10 min driving distance | 11.85% | 21.42% | 60.93% | 55.31% |

**Table A3.** *Cont.*

| Supply-A2*S2*N2 (Median) | Home Access Only | Workplace Access Only | Both Home and Workplace Access | Home Access/Workplace Access Ratio |
|---|---|---|---|---|
| ≤3 min walking distance | 0.26% | 6.17% | 0.01% | 4.13% |
| ≤5 min walking distance | 0.54% | 8.52% | 0.04% | 6.39% |
| ≤10 min walking distance | 1.54% | 12.57% | 0.21% | 12.28% |
| ≤3 min driving distance | 4.24% | 17.44% | 1.02% | 24.33% |
| ≤5 min driving distance | 11.85% | 26.78% | 6.28% | 44.26% |
| ≤10 min driving distance | 16.48% | 25.71% | 46.34% | 64.10% |
| **Supply-A3*S3*N3 (Strictest)** | **Home Access Only** | **Workplace Access Only** | **Both Home and Workplace Access** | **Home Access/Workplace Access Ratio** |
| ≤3 min walking distance | 0.09% | 1.67% | 0.00% | 5.26% |
| ≤5 min walking distance | 0.21% | 2.77% | 0.01% | 7.65% |
| ≤10 min walking distance | 0.62% | 6.06% | 0.04% | 10.22% |
| ≤3 min driving distance | 2.12% | 9.88% | 0.26% | 21.47% |
| ≤5 min driving distance | 7.92% | 18.67% | 2.19% | 42.43% |
| ≤10 min driving distance | 21.68% | 27.21% | 25.72% | 79.68% |

**Table A4.** Accessibility of supply-side opportunities for long-distance (≥30 miles) commuters.

| Supply-A1*S1*N1 (Best) | Home Access Only | Workplace Access Only | Both Home and Workplace Access | Home Access/Workplace Access Ratio |
|---|---|---|---|---|
| ≤3 min walking distance | 0.37% | 7.29% | 0.02% | 5.08% |
| ≤5 min walking distance | 0.81% | 10.10% | 0.06% | 8.00% |
| ≤10 min walking distance | 2.38% | 15.54% | 0.33% | 15.33% |
| ≤3 min driving distance | 6.02% | 21.37% | 1.59% | 28.16% |
| ≤5 min driving distance | 14.80% | 30.50% | 9.55% | 48.52% |
| ≤10 min driving distance | 12.66% | 23.28% | 58.29% | 54.39% |
| **Supply-A2*S2*N2 (Median)** | **Home Access Only** | **Workplace Access Only** | **Both Home and Workplace Access** | **Home Access/Workplace Access Ratio** |
| ≤3 min walking distance | 0.24% | 5.78% | 0.01% | 4.11% |
| ≤5 min walking distance | 0.52% | 7.91% | 0.03% | 6.62% |
| ≤10 min walking distance | 1.50% | 11.75% | 0.15% | 12.78% |
| ≤3 min driving distance | 4.21% | 16.69% | 0.80% | 25.24% |
| ≤5 min driving distance | 12.03% | 26.37% | 5.37% | 45.64% |
| ≤10 min driving distance | 17.32% | 27.35% | 43.49% | 63.33% |
| **Supply-A3*S3*N3 (Strictest)** | **Home Access Only** | **Workplace Access Only** | **Both Home and Workplace Access** | **Home Access/Workplace Access Ratio** |
| ≤3 min walking distance | 0.09% | 1.59% | 0.00% | 5.38% |
| ≤5 min walking distance | 0.21% | 2.60% | 0.00% | 8.12% |
| ≤10 min walking distance | 0.59% | 5.68% | 0.03% | 10.39% |
| ≤3 min driving distance | 2.04% | 9.41% | 0.21% | 21.66% |
| ≤5 min driving distance | 7.85% | 18.04% | 1.92% | 43.54% |
| ≤10 min driving distance | 22.10% | 27.91% | 23.75% | 79.18% |

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
