# Peer review of "Repurposing Existing Infrastructure for Urban Air Mobility: A Scenario Analysis in Southern California"

_drones, doi:10.3390/drones7010037_

Round 1

Reviewer 1 Report (Previous Reviewer 2)

Thank you for the opportunity to review the revised manuscript. The authors have addressed reviewer comments and the paper has improved greatly. 

Author Response

Thank you very much for your feedback. We wish you all the best in your research!

Reviewer 2 Report (Previous Reviewer 3)

Authors sufficiently addressed all comments from all reviewers 

Author Response

Thank you very much for your feedback. We wish you all the best in your research!

Reviewer 3 Report (New Reviewer)

This manuscript introduces a systematic approach how to identify suitable vertiport locations based on important criteria and constraints such as for example population characteristics (income, age, etc.), noise, restricted fly zones and addresses a very important challenge inside current UAM research and development. The area of southern California is indeed an interesting application area for UAM due to almost ideal weather conditions. 

However, this manuscript can improve its impact by giving a bit more background information about already existing research/project results for this area in the literature review section (e.g. Rimjha, M.; Hotle, S.; Trani, A.; Hinze, N.; Smith, J.C. Urban Air Mobility Demand Estimation for Airport Access: A Los Angeles International Airport Case Study or Rimjha, M.; Li, M.; Hinze, N.; Tarafdar, S.; Hotle, S.; Swingle, H.; Trani, A. Demand Forecast Model Development and Scenarios Generation for Urban Air Mobility Concepts; Virginia Tech Air Transportation Systems Laboratory 2020.).

In addition, and because European studies have already been introduced by the author, it may be worth having a look into the social acceptance study by EASA published in 2022 (uam-full-report.pdf (europa.eu)) 

In line 34 and throughout the manuscript, two different citation styles "[1]" and "[n1-n2]" were used. It is suggested to use only one citation style. 

In line 44, the quote refers to p.1 which causes confusion. If page 1 of the quote is important for context, please explain/refer to it in the manuscript. 

Table 1 summarizes the key insights of each listed source. It is suggested to rather provide a critical discussion about the individual outcomes in an "overarching" manner and highlight similarities and differences. 

In line 150, the author refers to Figure 1 which are only fictional renderings. The relation to Figure 1 is not clear. 

In line 161, the source of the 2018's American Community Survey is missing. Please add the correct reference.  

Figure 2 is not mentioned in the text. If there is no need for Figure 2 please remove it from the manuscript, otherwise it is suggested to provide a description and a crossrefence in the text. 

In line 211 the author provides the following statement: "The study defined three simple but crucial categories of rules [...]" Whereas in line 248 the author stated "As shown in Figure 3, five categories of supply-side rules [...]" It is suggested to further describe how many categories have been defined and why they might differ. 

Further background information is needed with respect to the categories se01, se02, se03. How are those income thresholds defined? Are they derived from the LEHD? 

Since the paragraph starting from line 323 is mainly describing Figure 5, it is recommended to directly refer to Figure 5 instead of using "(see Figure 5)". This makes it also easier to link the expression "Station X" to the subsequent figure.  

Author Response

Please find attached the reponse letter. Thanks.

This manuscript is a resubmission of an earlier submission. The following is a list of the peer review reports and author responses from that submission.

Round 1

Reviewer 1 Report

The author uses a novel data set for identifying the potential sites for UAM service; however, the author should work more on improving and reconstructing the quality of the article. Some apparent issues are as follows. 

Introduction section:

The description is verbose, with some complex and grammatically incorrect sentences.

Literature Review section:

It is unclear why the insights from the survey of GAV can be applied to UAM. Additionally, the 
"tradition aerial service" (line 77 on page 2) does not mean "on-demand aerial service." 

Data and analysis section:

"There are several reasons why I choose this area for our study" (line 146 on page 6) does not sound professional enough for a journal paper. 

The term, extreme weather conditions (line 152 on page 6), means different things than what the author would like to describe. 

The reviewer disagrees with the article's second point of selecting LA as the study area. The author should provide a more robust statement or references to support this point.  

The description of “demand coverage” in Table 3 is insufficient and unclear. 

Results:

The results only focus on the converges of OD points in the city with various sets of vertiports/airports; however, it does not include any evaluation of potential timesaving by utilizing the sets of vertiports/airports. For example, a long-distance commuter with a commute time of more than 1 hour can benefit from UAM by driving more than 10 minutes to the nearby airports/vertiports. 

The reason “commuters’ accessibility is not very sensitive to travel distance” (line 242 on page 10) is that the analysis does not compare the trip time with and without UAM service. So, the author’s statement and derivation are weak. 

Conclusion:

The conclusion is not well presented. And the author does not discuss how to improve the work in the second paragraph. 

Reviewer 2 Report

Thank you for the opportunity to review this scenario analysis to repurpose infrastructure to support urban air mobility in Southern California. A few questions and comments for the authors:

'Commercial UAM operations have begun in the 32 United States since at least the 1940s. For example, from 1947 to 1971, Los Angeles Air- 33 ways used helicopters to transport people and mail between dozens of locations in the 34 Los Angeles basin, including Disneyland and Los Angeles International Airport. During 35 the same era, from 1949 to 1979, the New York Airways primarily used helicopters to fly 36 people between helipads in Manhattan and airports in the New York area such as LaGuar- 37 dia, JFK, and Newark.'-Consider adding a citation for this history. Some resources are included below.

NASA conducted two market studies for urban air mobility. It seems a bit unusual that the authors did not reference or mention either of these studies. As such, the article and table 1 should include this literature as its foundational. Links to these studies are included below.

Some literature cites organizations (e.g., Airbus, Deloitte, etc.) where in fact there are author names. The authors should properly reference literature with author names whenever an author name is available.

What additional limitations of the methods employed or findings exist? Are there any recommendations for additional research?

Some additional resources that may be helpful for the authors:

https://journals.sagepub.com/doi/10.1177/03611981221076839

  https://nari.arc.nasa.gov/sites/default/files/attachments/UAM_ConOps_v1.0.pdf   https://www.semanticscholar.org/paper/Urban-Air-Mobility-Airspace-Integration-Concepts-Thipphavong-Apaza/c2d444f319b11b641e0b0f122e64c7b287e88310

https://www.mdpi.com/2071-1050/13/13/7421

https://ieeexplore.ieee.org/abstract/document/9447255   https://www.nexaadvisors.com/uam-global-markets-study   https://www.morganstanley.com/ideas/autonomous-aircraft

https://ntrs.nasa.gov/citations/20190001472

https://ntrs.nasa.gov/citations/20190002046

https://ntrs.nasa.gov/citations/20205007433   https://institutes.kpmg.us/content/dam/advisory/en/pdfs/2019/urban-air-mobility.pdf 

https://tsrc.berkeley.edu/publications/potential-societal-barriers-urban-air-mobility

https://www.easa.europa.eu/en/domains/urban-air-mobility-uam

https://www.semanticscholar.org/paper/An-Assessment-of-Public-Perception-of-Urban-Air-Yedavalli/f4641304ae7082a61d9cd82905917ce694a120c4

Reviewer 3 Report

The manuscript explores repurposing existing ground infrastructure for future aviation operations. In general the manuscript is well written. The literature review includes all the major relevant publications and I recommend a few more relevant references to strengthen the manuscript. The methodology is sound but some of the datasets and their use are questionable and some implicit assumptions are not well aligned with UAM concepts.
Urban air mobility (UAM) is an established term but the United States, primarily NASA, has favored the term advanced air mobility (AAM) as it's more general and not necessarily excludes non-urban operations, as UAM implies a focus only on urban locations. I don't think the manuscript needs to replace UAM for AAM throughout but it would be nice to recognize the AAM concept and it's relationship to UAM. Even adding advanced air mobility as a keyword should increase the discoverability of the manuscript.
The literature review is primarily peer-reviewed papers, which is good but I also recommend including two industry white papers: the Airbus and Boeing "New Digital Era of Aviation" and the Wisk and Skyport "Concept of Operations: Autonomous UAM Aircraft Operations and Vertiport Integration." These papers, particularly by Wisk and Skyport, strongly align with the discussion of the submitted manuscript.
There were two notable UAM papers by Vascik and Hansman that were not included in the review and I recommend their inclusion for completeness. They are the 2017 and 2018 Vascik and Hansman papers on operational constraints. Reference [20] by Vascik and Hansman is a good reference but it relies on the foundation established by the 2017 paper, which also assesses California like the submitted manuscript.
Vascik, P.D., Hansman, R.J. and Dunn, N.S., 2018. Analysis of urban air mobility operational constraints. Journal of Air Transportation, 26(4), pp.133-146
Vascik, P.D. and Hansman, R.J., 2017. Evaluation of key operational constraints affecting on-demand mobility for aviation in the Los Angeles basin: ground infrastructure, air traffic control and noise. In 17th AIAA Aviation Technology, Integration, and Operations Conference (p. 3084).
The manuscript includes references on demand modeling, like reference [15] by Fu et al. in 2019. I recommend adding two more papers. A 2021 paper by Rimjha et al. on a northern california use case that aligns with the manuscript's scope; and a 2021 paper by Alvarez et al. for a use case focused on New York City and which analyzed how vertiport locations influence operational metrics (and align with the recommended Vascik and Hansman papers)
Rimjha, M., Hotle, S., Trani, A. and Hinze, N., 2021. Commuter demand estimation and feasibility assessment for Urban Air Mobility in Northern California. Transportation Research Part A: Policy and Practice, 148, pp.506-524.
L. E. Alvarez, J. C. Jones, A. Bryan, and A. J. Weinert, “Demand and Capacity Modeling for Advanced Air Mobility,” in AIAA AVIATION 2021 FORUM, American Institute of Aeronautics and Astronautics. doi: 10.2514/6.2021-2381.
Reference [29] by Schweiger and Preis is an excellent systematic review. However it is referenced only on page 15. I strongly recommend citing it sooner in the manuscript in Section 2, along with some additional discussion about the contributions of it. Note all the references I recommended above were also recommended to Schweiger and Preis and I believe all of them are in it as well.
Table 1: the key findings of garrow et al is "study in progress." This is insufficient information and more information on the findings are required.
Figure 1: The figure is a bit confusing because only two of three archetypes are illustrated. I recommend having figure 1c as a standalone figure.
Line 142: Add introduction paragraph to each section
Line 144: Please add a simple figure illustrating the study area with the counties in scope. Readers should not be expected to know the country administrative boundaries. For more detail, this figure could also illustrate current airport locations, population density, ground transportation, or clusters of `super commuters`. Figure 4 is not comprehensive because it doesn't show the full counties and I'm unsure if that is the intent.
Line 159, please provide citation to LEHD data
Line 168, please provide citation to helipad data by FAA GeoHub. I'm very familiar of the FAA GeoHub by "helipads" are not a named dataset, do you mean the "airports" dataset (https://hub.arcgis.com/documents/f74df2ed82ba4440a2059e8dc2ec9a5d/explore)? The airport datasets assigns a facility type, such as heliport, to each airport record. I know from experience that there are many heliports that are recorded as operational but may never be used, such as a private heliport in someone's backyard. This is still a very good dataset to use but some discussion about the breadth and potential bias of using this dataset should be added to the manuscript.
Line 179: The UAS airspace and facility maps are not no fly zones. According to the FAA, UAS Facility Maps (UASFM) are “job aids used by FAA Part 107 processors to help them process airspace authorization requests. They depict the maximum altitude in feet AGL [sic] that may be assigned by a FAA processor without additional internal FAA coordination. UAS operators may use these altitudes as a guideline when submitting their UAS Airspace Authorization requests.” First, the UASFM were designed to support drone operations regulated by 14 CFR §107, which per 14 CFR § 107.31 require the drone operator to maintain visual line of sight of the drone. This manuscript assumes beyond visual line of sight operations of UAM aircraft for passengers onboard; both these manuscripts assumptions are not supported by 14 CFR §107.  Second, because the UASFM were not designed for BVLOS operations, the altitude limits of adjoining grids are not guaranteed to facilitate efficient or feasible transit through multiple grids. Third and most importantly, the UASFM are for airspace authorization requests only. You can still receive authorization to operate in a UASFM with 0 feet AGL ceiling, you just can't receive easy pre-approval using the UASFM. Table 3 correctly notes pre-approval altitudes, but this isn't sufficiently described in the text. The UASFM are also not a defined airspace class like Class B or D airspace, so be careful about describing UASM as "airspace zoning." Please update the manuscript accordingly to better describe the the intent of the UASFMs.
Section 4 and the discussion based on UASFM zoning: As noted in my previous comment the UASFM are tied directly to small drone line of sight operations. I am not aware of any UAM commuter concept of operations that have passenger carrying aircraft cruising below 400 feet. UAM aircraft at these very low altitudes would have an increased collision risk with small drones, which the manuscript does not discuss in detail, potentially be energy inefficient, and the noise generated by low flying aircraft would likely be unacceptable. Particularly the discussion on noise nuisances is only briefly mentioned in Section 4 and insufficiently discussed.
Table 2, please provide citations to various datasets
Line 314, the conclusion mentions "ease of certification" but this is the first instance that "certification" is used in the manuscript. The manuscript fails to cite standardization activities by ASTM, RTCA, ICAO, JARUS, and EuroControl to help facilitate certification; nor does the manuscript discuss air and ground risk. I recommend removing the mention of certification along with mention of ride quality and lifecycle emissions, as these are insufficiently to not all discussed in the manuscript.